# Relationship between Exercise Test Parameters, Device-Delivered Electric Shock and Adverse Clinical Events in Patients with an Implantable Cardioverter Defibrillator for Primary Prevention

**DOI:** 10.3390/jpm13040589

**Published:** 2023-03-28

**Authors:** Guillaume Théry, Laurent Faroux, Fanny Boyer, Pierre Nazeyrollas, Jean-Pierre Chabert, Damien Metz, François Lesaffre

**Affiliations:** 1Department of Cardiology, Reims University Hospital, 51100 Reims, France; 2Intensive Care Unit, Reims University Hospital, 51100 Reims, France

**Keywords:** exercise test, oxygen uptake, implantable cardioverter defibrillator, heart failure

## Abstract

(1) Background: Receiving the first internal electric shock is a turning point for patients with an implantable cardioverter defibrillator (ICD) for primary prevention. However, no study has investigated whether patients who receive a first device-delivered electric shock have a poor prognosis even at the time of ICD implantation. (2) Methods: We retrospectively identified 55 patients with ischemic (n = 31) or dilated (n = 24) cardiomyopathy who underwent ICD implantation for primary prevention with exercise test at the time of implantation. We recorded baseline characteristics, exercise test parameters, and clinical events. (3) Results: After a median follow-up of 5 years, we observed an association between an appropriate device-delivered electric shock, the occurrence of death or heart transplant, and the occurrence of the composite endpoint. There was also a significant relation between a VE/VCO_2_ slope >35 and the occurrence of the composite endpoint. Conversely, there was no significant association between negative outcomes on the exercise test and the occurrence of a device-delivered electric shock. (4) Conclusions: The exercise test performed at the time of ICD implantation do not predict the occurrence of device-delivered electric shock. The exercise test and the first electric shock are two independent markers of poor prognosis.

## 1. Introduction

Chronic heart failure (CHF) affects between 0.5 and 2% of the population in Europe and represents a major public health problem. Its prevalence is on the rise in developed countries due to population aging and thanks to improved management of ischemic heart disease, which is the primary cause of CHF [1]. Despite progress in therapeutic management, CHF continues to carry a poor prognosis, with mortality of up to 50% within the four years after diagnosis [2].

Therapeutic management of CHF in patients with a left ventricular ejection fraction (LVEF) < 40% is well described. In addition to specific treatment for the underlying cardiac pathology, first-line medical therapy includes renin–angiotensin system (RAS) antagonists (or alternatively, the association of sacubitril/valsartan), beta-blockers, and mineralocorticoid receptor antagonists. More recently, Sodium-glucose Cotransporter-2 (SGLT2) inhibitors have been shown to improve prognosis in CHF patients with LVEF < 40% [3].

A large proportion of deaths in patients with CHF are caused by ventricular arrhythmias [4]. Consequently, an implantable cardioverter defibrillator (ICD) for primary prevention is recommended for patients with an LVEF < 35% under optimal medical therapy [5,6,7]. It has previously been shown that the occurrence of device-delivered therapy was associated with increased mortality in these patients; Poole et al. showed in 2008 that among patients with heart failure in whom an ICD was implanted for primary prevention, those who received shocks for any arrhythmia (appropriate or not) had a substantially higher risk of death than similar patients who did not receive such shocks [8]. The idea that a first device-delivered shock constitutes a turning point in the course of disease is now widespread. However, we cannot rule out the possibility that patients who experience a first device-delivered shock actually had a poorer prognosis at the time of implantation already, and no studies have established the link between exercise-test parameters and device-delivered shock, while peak VO_2_ is closely related to mortality. Moreover, a modest increase in peak VO_2_ is associated with more favorable outcomes and decreased mortality [9]. Numerous facets of the relationship between the clinical status at the time of ICD implantation, the occurrence of device-delivered therapy, and subsequent clinical events remain to be elucidated.

We therefore sought to investigate the impact of the exercise test parameters at the time of ICD implantation on the occurrence of a first device-delivered shock as well as on clinical events. Secondly, we investigated the relation between device-delivered shock and clinical events (death or heart transplant, and a composite endpoint including death, heart transplant, hospitalization for acute heart failure, stroke or acute coronary syndrome).

## 2. Materials and Methods

### 2.1. Study Population

All patients with CHF and LVEF < 40% scheduled to receive implantation of an ICD for primary prevention, and who had an exercise test in the 6 months prior to implantation, at the University Hospital of Reims, France between 1 January 2012 and 1 August 2019 were retrospectively identified from the hospital informatics database (N = 66 patients). Patients with cardiomyopathy other than ischemic or dilated cardiomyopathy and those who had ICD implantation for secondary prevention were excluded (N = 11). The final study population therefore comprised 55 patients.

From the patients’ medical records, we recorded baseline characteristics, parameters of the exercise test, and follow-up data. All patients attended a follow-up consultation at our institution every 6 months and within 48 h if an electric shock was delivered by the device. The majority of patients had tele-monitoring of their device. All patients and their general practitioners were contacted by telephone to obtain information about clinical events occurring during follow-up. The clinical events recorded were death, heart transplant, hospitalization for acute heart failure, stroke or acute coronary syndrome (ACS).

All patients consented to the use of their anonymized medical data.

### 2.2. Exercise Test

All patients performed an exercise test on a cycle ergometer (General Electric eBike III, GE Healthcare, Chicago, IL, USA) with measurement of gaseous exchange (CPX, Viasys Healthcare, Yorba Linda, CA, USA). All exercise tests were performed in the presence of a cardiologist and a nurse or two cardiologists (a senior and a resident). After the test, we measured the weight and height of each patient. The first part of the exercise test comprised spirometric tests to determine the forced expiratory volume in 1 s (FEV1), expiratory flow at 25, 50 and 75% of vital capacity, and mean expiratory flow between 25 and 75% of vital capacity. Tests had to performed at a hygrometry level between 50 and 60% and at a temperature of between 20 and 25° Celsius. We recorded any symptoms occurring during exercise and requiring interruption of the test, heart rate (as monitored by 12-lead ECG), pulse oximetry, and blood pressure as measured by an armband. Patients rested for 5 min before beginning the exercise test to achieve baseline metabolic values and to attain a respiratory exchange ratio (RER), which is defined as the ratio of VCO_2_/VO_2_, between 0.75 and 0.85, which is indicative of a stable state. The 5-min rest period was then followed by 8 to 12 min of exercise test, following a standard ramp protocol [10] estimated in advance by calculating the maximum theoretical VO_2_ and the corresponding power in watts for each subject, so that the exercise would last on average 10 min. After the test, there was a recovery phase of 10 min. Patients were required to maintain a cadence of 60 ± 4 revs per minute on the cycle ergometer, and the test was considered to be interpretable and maximal if RER >1.10 and the patient had reached exhaustion. Patients were encouraged during the exercise and urged to continue until exhaustion.

Negative results on the exercise test included peak VO_2_ < 12 mL/min/kg, peak VO_2_ < 50% of the theoretical value, drop in pulse oximetry during exercise, circulatory power (defined as peak VO_2_ multiplied by peak systolic blood pressure) < 2000, presence of oscillatory ventilation, a VE/VCO_2_ slope > 35, and a drop in arterial blood pressure at the end of the exercise.

### 2.3. Electric Shocks Delivered by the Device and Clinical Events

Ventricular arrhythmia may be treated by antitachycardia pacing (ATP) or by an internal electric shock delivered by the implanted device. Current guidelines favor ATP [11], where possible, as it is less painful than internal electric shock delivery. ICD parameters were standardized, with a ventricular tachycardia zone of 170–180 beats per minute (bpm), and 220–230 bpm, to initiate one or more electric pulses, followed by an electric shock if the stimulation with shorter bursts of pulses failed; and a zone of ventricular fibrillation beyond 220–230 bpm, which initiates an internal electric shock with ATP before or during the charge. We recorded all therapies delivered by the ICD and obtained during monitoring of the device at follow-up, namely electric shock with or without ATP, and appropriate internal electric shock, with or without ATP. An internal electric shock delivered by the device was considered appropriate if it was delivered further to tachycardia or ventricular fibrillation confirmed by the data downloaded from the device.

The very first internal electric shock was suspected clinically and confirmed after ICD interrogation during an urgent medical consultation or tele-monitoring.

### 2.4. Statistical Analysis

Quantitative variables are expressed as mean ± standard deviation (SD) or median and interquartiles. Qualitative variables are expressed as number and percentage. Quantitative variables were compared using the Student’s t or Mann–Whitney U test, and qualitative variables were compared using the chi square or Fisher’s exact test, as appropriate. We constructed survival curves using the Kaplan–Meier method, and curves were compared using the log rank test. The factors associated with the occurrence of clinical events were identified using univariate Cox models. The impact of the delivery of a first electric shock by the device on the occurrence of clinical events was also studied using univariate Cox models, using the variable “delivery of a first electric shock” as a time-dependent variable. A *p*-value < 0.05 was considered statistically significant. All analyses were performed using SAS version 9.4 (SAS Institute Inc., Cary, NC, USA) and Prism version 8.1.2 (GraphPad Software, San Diego, CA, USA).

## 3. Results

### 3.1. Patient Characteristics

The characteristics of the 55 patients included in the study are presented in Table 1. The median age was 60 years (47, 73) and 75% were men. Median left ventricular ejection fraction (LVEF) was 30% (24, 36) and 85% of patients had NYHA class ≥ 3 dyspnea. The underlying cause was ischemic heart disease in 56% and non-ischemic dilated cardiomyopathy in the remaining 44%. All patients had an ICD for primary prevention, and 19/55 (35%) had associated cardiac resynchronization. Almost all (98%) patients were receiving a beta-blocker, and more than half had an angiotensin-converting enzyme inhibitor (ACEI) or angiotensin receptor blocker (ARB) (56%) or an association of sacubitril/valsartan (42%). Forty-one patients (75%) were treated with anti-aldosterone, and 11 (20%) were treated with long-term amiodarone. No patient was receiving a sodium–glucose cotransporter-2 (SGLT2) inhibitor, since our study was carried out before these drugs were introduced in the setting of heart failure.

### 3.2. Exercise Test Results

On exercise testing, 7 patients (13%) had a peak VO_2_ < 12 mL/kg/min, and 12 (22%) had a peak VO_2_ < 50% of the theoretical value. Almost half the patients had a VE/VCO_2_ slope > 35 (48%). Overall, 15 patients (33%) had circulatory power < 2000, 23 (45%) had a drop in pulse oximetry during exercise, and 25 (46%) had a drop in blood pressure during exercise. The other parameters recorded during exercise testing are presented in the Appendix A. During follow-up (median duration 4.9 years (3.9, 5.7)), five patients (9%) received an appropriate electric shock delivered by their device, eight patients (14%) had a non-appropriate electric shock, and ATP was delivered to seven patients (13%). Seven patients died (13%), and one underwent a heart transplant. The composite endpoint occurred in 17 patients (31%).

### 3.3. Association between Exercise Test Variables and Clinical Events

The negative findings on the exercise test included peak VO_2_ < 12 mL/min/kg, peak VO_2_ < 50% of the theoretical value, drop in pulse oximetry during exercise, circulatory power (defined as peak VO_2_ multiplied by peak systolic blood pressure) < 2000, presence of oscillatory ventilation, and a drop in arterial blood pressure at the end of the exercise. The details are provided in Table 2. None of these criteria was associated with the occurrence of death or heart transplant. Conversely, a VE/VCO_2_ slope >35 was significantly associated with the occurrence of the composite endpoint of death, heart transplant, hospitalization for acute heart failure, stroke or acute coronary syndrome (hazard ratio (HR) 3.28, 95% confidence interval (CI) 1.15–9.41, *p* = 0.027) (Figure 1). None of the other factors was associated with the composite endpoint.

### 3.4. Association between Device-Delivered Electric Shock and Clinical Events

The impact of a device-delivered electric shock on clinical events in detailed in Table 3. The delivery of an appropriate electric shock was significantly associated with the occurrence of death or heart transplant (HR 10.11, 95%CI 1.66–61.52, *p* = 0.012) and with the occurrence of the composite endpoint (HR 5.39, 95%CI 1.29–22.50, *p* = 0.021) (Figure 2). Conversely, there was no relationship between the delivery of an electric shock, the delivery of an electric shock or ATP, or the delivery of an electric appropriate electric shock or ATP, and the occurrence of death, heart transplant or the composite endpoint.

### 3.5. Association between Exercise Test Variables and Appropriate Electric Shock

Table 4 presents the comparison of characteristics between patients who received an appropriate electric shock delivered by the device and those who did not. None of the characteristics studied was associated with the delivery of an appropriate electric shock. The type of cardiomyopathy (ischemic or dilated) was also unrelated to the delivery of an appropriate shock. Among the long-term treatments, the use of amiodarone was associated with the delivery of an appropriate electric shock (*p* = 0.004). None of the exercise test variables was related to delivery of an appropriate electric shock.

## 4. Discussion

To the best of our knowledge, this is the first study to assess the relationship between findings on the exercise test performed prior to ICD implantation and the occurrence of device-delivered electric shock or adverse clinical events. Our main findings are as follows: (1) a VE/VCO_2_ slope >35 was associated with the occurrence of the composite endpoint; (2) the delivery of an appropriate electric shock by the device was associated with the occurrence of death or heart transplant and with the composite endpoint; and (3) the baseline characteristics, including the exercise test parameters of patients who received an appropriate, device-delivered electric shock did not differ from those of patients who received no shock.

The association observed here between a VE/VCO_2_ slope >35 and the occurrence of the composite endpoint underscores the value of this parameter as a predictor of poor prognosis [12]; indeed, it is one that is often considered superior to peak VO_2_ [13,14]. The VE/VCO_2_ gradient reflects the ventilator response to exercise and the production of CO_2_. It depends on chemoreceptors, the pulmonary dead space, and also on the muscle mass engaged in producing the effort. However, the relation between VE and VCO_2_ is not linear, and VE increases disproportionately compared to VCO_2_ at the end of exercise, when the plasma pH decreases, and it is also linked to a drop in pulmonary perfusion that leads to ventilation–perfusion mismatch. A high VE/VCO_2_ slope is the reflection of unfavorable cardio-circulatory conditions, and this is reflected in its association with adverse outcomes via the composite endpoint. However, VE/VCO_2_ slope was not found to be associated with the harder endpoint comprising death or heart transplant, although this may be due to the small sample size of our study. In their meta-analysis, Poggio et al. [15] VE/VCO_2_ slope was associated with serious cardiovascular events, such as death, ventricular assist device implantation, or heart transplant. Association between VE/VCO_2_ slope and poor prognosis is well described in heart failure with reduced LVEF; in case of severe left ventricular systolic dysfunction (ischemic and non-ischemic cardiomyopathy), VE/VCO_2_ slope might be a non-invasive marker of advanced right ventricular dysfunction [16]. However, it might not be a good reflect of poor outcomes when LVEF is preserved [17].

In our study, the delivery of an appropriate electric shock by the device (and considered as a time-dependent variable) was found to be associated with the occurrence of events and confirms the fact that the first electric shock represents a turning point in the progression of the disease. In a substudy of the MADIT-II trial [18], patients who received successful appropriate therapy by an ICD for ventricular tachycardia or ventricular fibrillation were at increased risk of adverse outcomes with an increased risk of death. However, all the patients in that study had ischemic cardiomyopathy. This naturally raises the hypothesis that the delivery of ICD therapy is, in itself, an event that may worsen underlying cardiomyopathy. Although lifesaving, ICD therapy has side effects including changes to contractile function and myocardial relaxation [19], a reduction in cardiac output [20], an increase in the pacing threshold [21], and above all, irreversible tissue damage [22]. This tissue injury can increase the risk of developing heart failure, or it may compound symptoms [23]. Numerous observational and/or retrospective studies have investigated these effects, but none was designed to ascertain causality. Semmler et al. reported an increase in high-sensitivity troponin after intraoperative ICD testing using shock applications [24]. However, in those previous studies, every shock was harmful, but we found different results if the shock was appropriate or not.

The lack of association between delivery of a shock (whether appropriate or not) and the occurrence of adverse clinical events is in line with previous reports in the literature [24,25], and it could be explained by the multiple possible etiologies of inappropriate ICD therapies, including supraventricular tachycardia or myopotentials. Powell et al. [26] reported that the adverse prognosis observed after a first shock appears to be more related to the underlying arrhythmia than to any adverse effect of the shock itself. Accordingly, in their study, patients who received their first shock for atrial flutter/fibrillation or ventricular tachycardia/ventricular fibrillation had lower 3-year survival that those who received shocks for supraventricular tachycardia/sinus tachycardia or for noise/artefact/oversensing. Conversely, in the ADVANCE III study [24], which compared a standard to a long detection interval, the longer detection interval was found to reduce ATP and shocks, notably inappropriate shock delivery, albeit without any impact on mortality.

Our study did not show any relationship between negative outcomes on the exercise test and the occurrence of an appropriate device-delivered shock. However, this statement has to be nuanced regarding the very low rate of appropriate shocks, and it can be the results of chance. Taken together with the association between VE/VCO_2_ slope > 35 and the composite endpoint, and the impact of an appropriate shock on adverse clinical events, these findings support the hypothesis that the exercise test may be a marker of poor prognosis at the time of ICD implantation, but the first occurrence of device-delivered shock marks a turning point in the prognosis. These two markers seem to be independent of each other, and the findings on the exercise test at the time of ICD implantation do not appear to predict the later delivery of a shock by the device. In reality, the exercise test evaluates the full spectrum of the metabolic pathway, encompassing cardiac, ventilatory and muscle function. In this regard, the exercise test is likely a better evaluation of cardiocirculatory function than cardiac load or the risk of arrhythmia stemming from the cardiomyopathy. The analysis of other parameters could also have been informative, such as resting heart rate. Indeed, Calé et al. [27] reported in a prospective study of 61 heart failure patients with an ICD that resting heart rate and peak VO_2_ were both good independent predictors of arrhythmic events by multivariate analysis. However, the population of their study was more heterogeneous, with some patients having ICD for secondary prevention.

In our study, patients treated with amiodarone at the time of ICD implantation more frequently received an appropriate shock. One of the main indications for the amiodarone treatment was episodes of atrial fibrillation. However, this result could suggest that the treating physician had previously identified episodes of ventricular extrasystoles or unsustained ventricular tachycardia, justifying the initiation of amiodarone. Overall, these patients were already at increased risk of presenting sustained ventricular arrhythmia. Thus, the existence of unsustained ventricular arrhythmias prior to the ICD implantation, which did not lead to ICD implantation as secondary prevention, could be more informative than exercise testing with regard to the risk of arrhythmia and the associated probability of receiving an electric shock, associated with poorer prognosis. This is in line with the idea that one risk factor for ventricular tachycardia is having had at least one episode of it in the past.

Finally, the prediction of device-delivered electric shock could be based on several parameters [28], including clinical variables (e.g., NYHA class), echographic findings (LVEF), and also biology (NT-proBNP or glomerular filtration rate). Estimating the risk of arrhythmia in this way, with the aid of other techniques such as cardiac magnetic resonance imaging (MRI) to quantify myocardial fibrosis, is likely highly informative and could help to predict sustained ventricular arrhythmias [29]. Morphological analysis of ventricular arrhythmias from the data stored on the ICD could also be of value in estimating patient risk.

### Study Limitations

Our study suffers from some limitations that are inherent to retrospective studies. Firstly, the sample size was quite small and the stress tests were not carried out prospectively and systematically. They were made as part of clinical management and could therefore induce a selection bias. The selected HF population was particularly young, with a high rate of NYHA class ≥III, making them potentially eligible for a transplant. We know that patients with severely symptomatic heart failure have an increased risk of dying from end-stage heart failure and not from rhythmic events, so the selected population does not represent a standard population of patients eligible for primary implantation of an ICD. A comparison of the study population to patients implanted during the study period and who did not undergo a stress test could be an issue for a further study. However, we had long follow-up information for the population (median 5 years). We did not use any other methods for evaluating the risk of arrhythmia, which might have provided additional insights, e.g., cardiac MRI to quantify myocardial fibrosis [30], or biological parameters (lactates, ammonia), which reflect hemodynamic adaptation. These would be interesting avenues for future research.

## 5. Conclusions

Our study shows that a first electric shock delivered by an ICD was associated with the occurrence of adverse clinical events in patients with ICD for primary prevention. The data from the exercise test at the time of implantation did not make it possible to predict the occurrence of an appropriate device-delivered shock. Our results suggest that the exercise test and the first device-delivered shock are two independent markers of poor prognosis that carry distinct information. Further prospective studies in larger populations are warranted to explore the relationships between the exercise test parameters, biological findings, myocardial fibrosis, the first device-delivered shock, and clinical outcomes in this population.

## Figures and Tables

**Figure 1 jpm-13-00589-f001:**
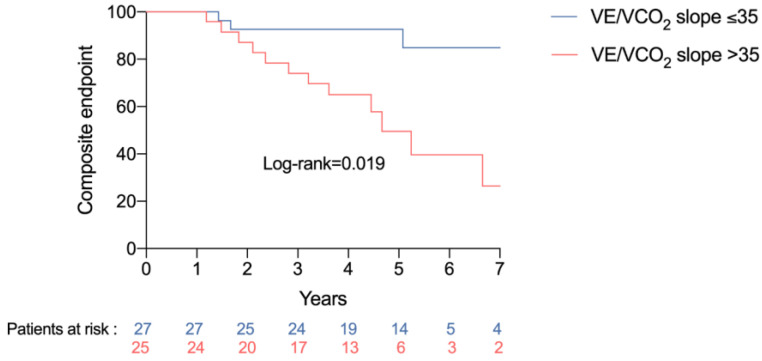
Occurrence of the composite endpoint according to VE/VCO_2_ slope.

**Figure 2 jpm-13-00589-f002:**
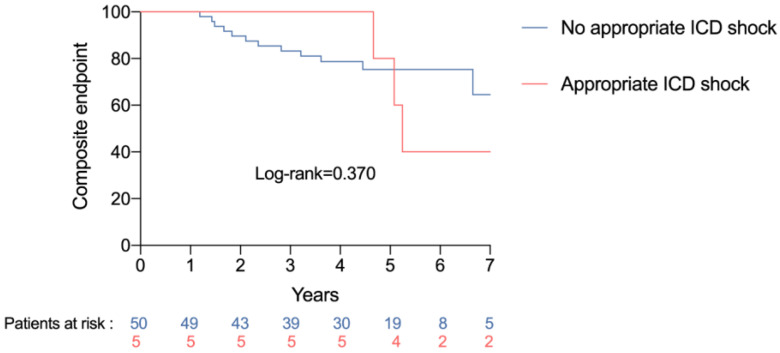
Occurrence of the composite endpoint according to appropriate device-delivered electric shock.

**Table 1 jpm-13-00589-t001:** Baseline characteristics of the study population and events during follow-up.

Baseline characteristics:	n = 55
Age, years	60 (47–73)
Male sex	41 (75)
NYHA class ≥ 3	47 (85)
LVEF, %	30 (24–36)
Dyslipidemia	27 (49)
Diabetes	19 (35)
Active smoking	36 (65)
Hypertension	17 (31)
Obesity	14 (25)
Atrial fibrillation	10 (18)
GFR < 60 mL/min	20 (36)
Stroke	2 (4)
COPD	4 (8)
Hyperthyroidism	4 (8)
Anemia	11 (20)
Obstructive sleep apnea	6 (11)
Underlying cardiomyopathy:	
Ischemic	31 (56)
Dilated	24 (44)
Treatment:	
ACEI/ARB	31 (56)
Sacubitril/Valsartan	23 (42)
Beta-blocker	54 (98)
Anti-aldosterone agent	41 (75)
Amiodarone	11 (20)
Loop diuretic	33 (60)
Ivabradine	15 (27)
Type of device implanted:	
Single chamber	24 (43)
Dual chamber	12 (22)
Cardiac resynchronization device	19 (35)
Negative outcomes on exercise test:	
Peak VO_2_ < 12 mL/min/kg	7 (13)
Maximum theoretical VO_2_ < 50%	12 (22)
VE/VCO_2_ slope > 35	25 (48)
Oscillatory ventilation	3 (7)
Circulatory power < 2000	15 (33)
Drop in pulse oximetry during exercise	23 (45)
Drop in blood pressure during exercise	25 (46)
Clinical events:	
Appropriate DDES	5 (9)
Appropriate shock or ATP	14 (25)
Death or heart transplant	8 (15)
Composite endpoint *	17 (31)
Median follow-up duration, months	59 (47–68)
Median time between ICD implantation and first DDES, months	37 (13–61)

ATP, antitachycardia pacing; DDES, device-delivered electric shock; NYHA, New York Heart Association; LVEF, left ventricular ejection fraction; GFR, glomerular filtration rate; COPD, chronic obstructive pulmonary disease; ACEI, angiotensin-converting enzyme inhibitor; ARB, angiotensin receptor blocker. * Composite of death, heart transplant, hospitalization for acute heart failure, stroke or acute coronary syndrome.

**Table 2 jpm-13-00589-t002:** Negative outcomes on exercise test and clinical events.

	Death or Heart Transplant		Composite Endpoint *	
	HR (95%CI)	*p*-Value	HR (95%CI)	*p*-Value
Peak VO_2_ < 12 mL/min/kg	0.60 (0.07–5.14)	0.640	1.73 (0.55–5.43)	0.350
Maximum theoretical VO_2_ < 50%	0.89 (0.17–4.55)	0.893	2.04 (0.75–5.56)	0.163
VE/VCO_2_ slope > 35	3.36 (0.67–16.80)	0.140	3.28 (1.15–9.41)	0.027
Oscillatory ventilation **	-	-	-	-
Circulatory power < 2000	1.25 (0.28–5.52)	0.765	1.65 (0.59–4.65)	0.341
Drop in pulse oximetry during exercise	0.92 (0.23–3.71)	0.904	0.70 (0.27–1.86)	0.480
Drop in blood pressure during exercise	1.80 (0.43–7.61)	0.423	1.40 (0.54–3.66)	0.488
Any two or more criteria	5.56 (0.68–46)	0.108	2.27 (0.78–6.54)	0.130

HR, hazard ratio; 95%CI, 95% confidence interval. * Composite of death, heart transplant, hospitalization for acute heart failure, stroke or acute coronary syndrome. ** The 3 patients with oscillatory ventilation had no adverse clinical events.

**Table 3 jpm-13-00589-t003:** Device-delivered electric shock and clinical events.

	Death or Heart Transplant		Composite Endpoint *	
	HR (95%CI)	*p*-Value	HR (95%CI)	*p*-Value
Electric shock	3.25 (0.69–15.21)	0.135	2.58 (0.78–8.53)	0.121
Shock or antitachycardia pacing	2.43 (0.52–11.36)	0.259	1.97 (0.61–6.41)	0.260
Appropriate shock	10.11 (1.66–61.52)	0.012	5.39 (1.29–22.50)	0.021
Appropriate shock or antitachycardia pacing	2.43 (0.52–11.36)	0.259	1.97 (0.61–6.41)	0.260

* Composite of death, heart transplant, hospitalization for acute heart failure, stroke or acute coronary syndrome.

**Table 4 jpm-13-00589-t004:** Comparison of the population characteristics between those who received and those who did not received an appropriate device-delivered electric shock.

	Shock (n = 5)	No Shock (n = 50)	*p*-Value
Baseline characteristics:			
Age, years	59 (55–60)	61 (47–68)	0.703
Men	5 (100)	36 (72)	0.314
NYHA Class ≥ 3	0	8 (16)	1.000
LVEF, %	30 (30–33)	30 (25–30)	0.276
Dyslipidemia	3 (60)	24 (48)	0.670
Diabetes	0	19 (38)	0.152
Active smoking	4 (80)	32 (64)	0.650
Hypertension	0	17 (34)	0.310
Obesity	0	14 (28)	0.314
Atrial fibrillation	1 (20)	9 (18)	1.000
GFR < 60 mL/min	2 (40)	18 (36)	1.000
Stroke	0	2 (4)	1.000
COPD	0	4 (8)	1.000
Hyperthyroidism	0	4 (8)	1.000
Anemia	0	11 (22)	0.571
Obstructive sleep apnea	0	6 (12)	1.000
Underlying cardiomyopathy:			
Ischemic	2 (40)	29 (58)	0.643
Dilated	3 (60)	21 (42)	
Treatment:			
ACE/ARB	2 (40)	29 (58)	0.643
Sacubitril/Valsartan	3 (60)	20 (40)	0.639
Beta-blocker	5 (100)	49 (98)	1.000
Anti-aldosterone agent	4 (80)	37 (74)	1.000
Amiodarone	4 (80)	7 (14)	0.004
Loop diuretic	4 (80)	29 (58)	0.638
Ivabradine	5 (100)	35 (70)	0.308
Type of device implanted:			
Single chamber	4 (80)	20 (40)	-
Dual chamber	0	12 (24)	-
Cardiac resynchronization device	1 (20)	18 (36)	0.649
Negative outcomes on exercise test:			
Peak VO_2_ < 12 mL/min/kg	0	7 (14)	1.000
Maximum theoretical VO_2_ < 50%	1 (20)	11 (22)	1.000
VE/VCO_2_ slope > 35	2 (40)	23 (46)	1.000
Oscillatory ventilation	0	3 (6)	1.000
Circulatory power < 2000	1 (20)	14 (28)	1.000
Drop in pulse oximetry during exercise	3 (60)	20 (40)	0.647
Drop in blood pressure during exercise	2 (40)	23 (46)	1.000
Any two negative outcomes	3 (60)	26 (52)	1.000
Any three negative outcomes	2 (40)	16 (32)	1.000

NYHA, New York Heart Association; LVEF, left ventricular ejection fraction; GFR, glomerular filtration rate; COPD, chronic obstructive pulmonary disease; ACEI, angiotensin-converting enzyme inhibitor; ARB, angiotensin receptor blocker.

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
