# Peer review of "Relationship between Exercise Test Parameters, Device-Delivered Electric Shock and Adverse Clinical Events in Patients with an Implantable Cardioverter Defibrillator for Primary Prevention"

_jpm, 2023, doi:10.3390/jpm13040589_

Round 1

Reviewer 1 Report

I review the paper very carefully because this is an interesting topic. The methodology, the results and the discussion are great. It is important for the readers to learn from these experiences. 

One simple question: WHY is there a need for 2 cardiologists to do the stress test?  All exercise tests were performed in the presence of a cardiologist and a nurse, or two cardiologists.

Author Response

We thank the reviewer for the insightful review and comments.
We answered directly to the comment from the reviewer by changing our manuscript. Corresponding changes to the revised manuscript are marked in copy of the manuscript.

1/ Why is there a need for 2 cardiologists to do the stress test?

Modification has been made. When stress test was done by two cardiologists, there were a senior and a resident.

Reviewer 2 Report

Most of the disadvantages of the manuscript and the results are stated in Limitations.

Introduction is not informative. The authors stated there are numerous studies, but conflicting results/studies and conclusions are not presented. These should be stated in this sections in order to support the aim. 

In methodology, endpoints are called clinical events, and then mentioned as composite endpoints. Please clarify this and harmonize throughout the text.

Discussion is large and ambiguous. Pleas focus on the results and their relevance to other studies, particularly VE/VCO2 in relation to the endpoints.

If the study is retrospective analysis of anonymized medical history, why the participants have Informed consent forms? Was it a local requirement in addition to the Ethics committee. Please provide Ethics committee approval details. 

References should be updated in terms of adding some recent similar studies. Also, the references should be written according to the instructions of the authors.

Author Response

We thank the reviewer for the insightful review and comments.
We answered directly to the comment from the reviewer by changing our manuscript. Corresponding changes to the revised manuscript are marked in copy of the manuscript.

1/ Introduction is not informative. The authors stated there are numerous studies, but conflicting results/studies and conclusions are not presented. These should be stated in this sections in order to support the aim. 

The modification has been made.

2/ In methodology, endpoints are called clinical events, and then mentioned as composite endpoints. Please clarify this and harmonize throughout the text.

The modification has been made.

3/ Discussion is large and ambiguous. Please focus on the results and their relevance to other studies, particularly VE/VCO2 in relation to the endpoints.

References have been added concerning VE/VCO2 slope and outcomes. The discussion section has been completed. 

4/ If the study is retrospective analysis of anonymized medical history, why the participants have Informed consent forms? Was it a local requirement in addition to the Ethics committee. Please provide Ethics committee approval details. 

The University Hospital of Reims has no Ethics comittee. When patients are hospitalized, then consent to provide personal datas (anonymously) for retrospective studies. 

5/ References should be updated in terms of adding some recent similar studies. Also, the references should be written according to the instructions of the authors.

The modification has been made.

Reviewer 3 Report

The study by Théry et al. investigates the relationship between exercise test parameters, device-delivered electric shock, and adverse clinical events in patients with an implantable cardioverter defibrillator (ICD) for primary prevention. However, there are so many aspects of the study, which require further clarification.

Only few of such comments are the following:

Firstly, the authors should provide an exact definition of the first internal electric shock with regard to the time when it occurs after ICD implantation. It is unclear from the study when the first internal electric shock occurs, and this information could affect the interpretation of the results. Additionally, the authors should describe in detail how they recorded and documented the first internal electric shock.

Secondly, the comparison of only five patients who experienced the first internal electric shock with 50 patients who did not experience the first internal electric shock may not be appropriate. This small sample size could lead to chance findings, and the results derived from such a comparison may not be valid. Therefore, the authors should avoid such comparison method and take completely different directions to rewrite their manuscript.

Thirdly, the authors should clarify whether any patients experienced the first internal electric shock during the follow-up period of 7 years. This information could provide additional insights into the relationship between the first internal electric shock and adverse clinical events.

Fourthly, the authors state that receiving the first internal electric shock is a turning point for patients with an ICD for primary prevention. However, they should provide a basis for this claim with appropriate citations from the literature. Why should it be a turning point while no one is calling it or proving it as a turning point? Very confusing terminology!!!

Fifthly, the authors state that the exercise was done at the time of ICD implantation, while the methods section state that the exercise was done within and prior to 6 months before ICD implantation. This discrepancy is very confusing and misleading.

Sixth, the title "Relationship between exercise test parameters, device-delivered electric shock and adverse clinical events in patients with an implantable cardioverter defibrillator for primary prevention" may be misleading as it may indicate that ICD does not play a role in primary prevention. Here, the authors presenting the function of ICD as a factor of adverse outcome, which is also very dangerous in terms of therapeutic utility of ICD.

Finally, the based on the unscientific and unjustified aims of the study, I would suggest authors just be very precise with the exact definitions of the terminology and completely rewrite the manuscript – I would suggest that they just compare between ischemic (n=31) or dilated (n=24) cardiomyopathy patients or just take the whole patient cohort and provide descriptive writing (information).

In conclusion, the many aspects of the study including definitions, terms and aims require further clarification to ensure the validity and interpretation of the results and to avoid confusions.

Author Response

We thank the reviewer for the insightful review and comments.
We answered directly to the comment from the reviewer by changing our manuscript. Corresponding changes to the revised manuscript are marked in copy of the manuscript. The all text has been read and corrected by an English native speaker.

1/ Firstly, the authors should provide an exact definition of the first internal electric shock with regard to the time when it occurs after ICD implantation. It is unclear from the study when the first internal electric shock occurs, and this information could affect the interpretation of the results. Additionally, the authors should describe in detail how they recorded and documented the first internal electric shock.

Precisions and modifications have been made.

2/ Secondly, the comparison of only five patients who experienced the first internal electric shock with 50 patients who did not experience the first internal electric shock may not be appropriate. This small sample size could lead to chance findings, and the results derived from such a comparison may not be valid. Therefore, the authors should avoid such comparison method and take completely different directions to rewrite their manuscript.

We took in consideration the reviewer's comment and precisions in the discussion section have been made.

3/ Thirdly, the authors should clarify whether any patients experienced the first internal electric shock during the follow-up period of 7 years. This information could provide additional insights into the relationship between the first internal electric shock and adverse clinical events.

Modification have been made and information added in Table 1.

4/ Fourthly, the authors state that receiving the first internal electric shock is a turning point for patients with an ICD for primary prevention. However, they should provide a basis for this claim with appropriate citations from the literature. Why should it be a turning point while no one is calling it or proving it as a turning point? Very confusing terminology!!!

Precisions have been made, developing Poole et al results (Poole, N Engl J of Med, 2008) that stated among 811 patients with heart failure in whom an ICD was implanted for primary prevention, those who received shocks for any arrhythmia had a substantially higher risk of death than similar patients who did not receive such shocks, with a median follow-up of 45 months. Mortality increased notably after the first device-delivered electric shock. The use of the expression "turning point" illustrates this statement.

5/ Fifthly, the authors state that the exercise was done at the time of ICD implantation, while the methods section state that the exercise was done within and prior to 6 months before ICD implantation. This discrepancy is very confusing and misleading.

Precisions have been made. Given the median time of follow-up and fatal arrythmias occurence in ischemic and dilated cardiomyopathies described in literature, a 6-month period between ICD implantation and exercice test seemed to be acceptable. But it is true that it's a notable bias in our study, part of retrospective data collection.

6/ Sixth, the title "Relationship between exercise test parameters, device-delivered electric shock and adverse clinical events in patients with an implantable cardioverter defibrillator for primary prevention" may be misleading as it may indicate that ICD does not play a role in primary prevention. Here, the authors presenting the function of ICD as a factor of adverse outcome, which is also very dangerous in terms of therapeutic utility of ICD.

We took in consideration the reviewer's comment. We briefly noticed in the discussion section that ICD therapeutic could worsen the disease course but this hypothesis is clearly not proven. We underligned that ICD implantation changed heart failure management and reduced mortality following international guidelines. 

Round 2

Reviewer 2 Report

Thank you. No further questions.

Author Response

Thank you

Reviewer 3 Report

ok

Author Response

Thank you